# Mapping Uncertainties of Soft-Sensors Based on Deep Feedforward Neural Networks through a Novel Monte Carlo Uncertainties Training Process

Erbet A. Costa [1], Carine M. Rebello [2,3], Vinicius V. Santana [2], Alírio E. Rodrigues [2], Ana M. Ribeiro [2], Leizer Schnitman [1] and Idelfonso B. R. Nogueira [2,*]

[1] Programa de Pós-Graduação em Mecatrônica, Escola Politécnica (Polytechnic School), Universidade Federal da Bahia, Salvador 40210-630, Brazil; erbetcosta@ufba.br (E.A.C.); leizer@ufba.br (L.S.)

[2] Laboratory of Separation and Reaction Engineering, Associate Laboratory LSRE/LCM, Department of Chemical Engineering, Faculty of Engineering, University of Porto, Rua Dr. Roberto Frias, 4200-465 Porto, Portugal; carine.menezes@ufba.br (C.M.R.); up201700649@edu.fe.up.pt (V.V.S.); arodrig@fe.up.pt (A.E.R.); apeixoto@fe.up.pt (A.M.R.)

[3] Programa de Pós-Graduação em Engenharia Industrial, Escola Politécnica (Polytechnic School), Universidade Federal da Bahia, Salvador 40210-630, Brazil

* Correspondence: idelfonso@fe.up.pt

**Abstract:** Data-driven sensors are techniques capable of providing real-time information of unmeasured variables based on instrument measurements. They are valuable tools in several engineering fields, from car automation to chemical processes. However, they are subject to several sources of uncertainty, and in this way, they need to be able to deal with uncertainties. A way to deal with this problem is by using soft sensors and evaluating their uncertainties. On the other hand, the advent of deep learning (DL) has been providing a powerful tool for the field of data-driven modeling. The DL presents a potential to improve the soft sensor reliability. However, the uncertainty identification of the soft sensors model is a known issue in the literature. In this scenario, this work presents a strategy to identify the uncertainty of DL models prediction based on a novel Monte Carlo uncertainties training strategy. The proposed methodology is applied to identify a Soft Sensor to provide a real-time prediction of the productivity of a chemical process. The results demonstrate that the proposed methodology can yield a soft sensor based on DL that provides reliable predictions, with precision being proven by its corresponding coverage region.

**Keywords:** soft sensor; deep feedforward neural network; uncertainty evaluation

## 1. Introduction

A usual problem found in engineering is the measurement of unmeasurable quantities. This motivates many studies in the literature addressing this problem [1–5]. The usual approach to address this problem is the development of machine learning-based (ML) models to perform the state predictions in real-time. This strategy has been providing solutions to several types of problems. For instance, in Nogueira et al., 2017 [4], the authors have developed a soft sensor based on artificial neural networks to predict the melt flow index of the polymers produced in an industrial application. In Capriglione et al., 2017 [6], the authors propose a soft sensor to provide real-time measurement of rear suspension stroke in two-wheeled vehicles.

A soft sensor is a well-known strategy to obtain information about a variable that is difficult or economically expensive to measure [5,7]. It is based on a mathematical model that can relate a set of variables to another set that interferes with them [8]. The soft sensor can be an economical measurement alternative [4]. When the measurement requires sophisticated techniques or expensive instruments, this strategy is presented as a reliable alternative. Even though it is a field that has been explored for several

years, this is still an open research area. For instance, the synergy between soft sensors and the recent advances in artificial intelligence (AI) is an issue that requires attention. An example of this is the development of deep learning techniques that have shown the potential to represent with precision several data-driven systems [7–10]. However, deep learning-based models present a drawback: an insufficient capacity to deal with uncertain scenarios [11].

According to the Guide to the Expression of Uncertainty in Measurements (GUM) by the Bureau International des Poids et Mesures (BIPM) [12], uncertainty is "a parameter, associated with the result of a measurement, that characterizes the dispersion of the values that could reasonably be attributed to the measurand.". When related to models' prediction, uncertainty is defined as a parameter representing the dispersion associated with a model's prediction. Thus, imperfections, hypotheses, and idealizations imposed during building a model contribute to its uncertainty [2]. The soft sensor literature has pointed out the issue of forecast uncertainty as an unresolved issue in this field [13].

Since these sensors are applied in a scenario where their inputs are subject to uncertainties and several other problems related to the data acquisition, they need to assess these uncertainties. On the other hand, as Gal et al. 2015 [14] argued, machine learning models do not capture uncertainties. Addressing this issue at the deep learning level allows the reliable application of these techniques in this field. For instance, Gal et al. [14] proposed a Bayesian approximation technique to assess uncertainty in deep learning. This is considered a seminal work in this field, providing the basis for a better understanding of the uncertainty in deep reinforcement learning. However, fully understanding the uncertainty of ML models is still a complex issue as it means taking the uncertainty of predictions. In this sense, Abdar et al. 2021 [15] present an in-depth literature review on uncertainty analysis in deep learning models and indicates several methods capable of evaluating the uncertainties of these models. Le et al. 2021 [11] addressed this problem by proposing an uncertainty-aware soft sensor based on a Bayesian deep recurrent neural network. The authors propose a contribution for studies addressing the uncertainty evaluation of deep learning techniques and their application on soft sensors. However, few works in the literature address this topic from the sensor perspective. The other works available address the uncertainty from the measured variables (soft sensor inputs) perspectives [16,17]. Hence, there is a lack of further investigations on bringing the uncertainty of the DL model in their application context. This is the main contribution of this work; it presents a methodology capable of providing a comprehensive view of these problems.

In this context, the present work proposes a novel strategy for uncertainty evaluation of deep feedforward neural networks based on a Monte Carlo uncertainty training. The proposed methodology is applied to evaluate the prediction uncertainty of a soft sensor developed to provide a real-time prediction of chemical process productivity. The syngas purification through a pressure swing adsorption unit is a case study due to its complex dynamics.

## 2. Methodology

Figure 1 depicts the proposed methodology. It comprises four steps to obtain a soft sensor and its corresponding uncertainty. This section will provide an overall description of each step.

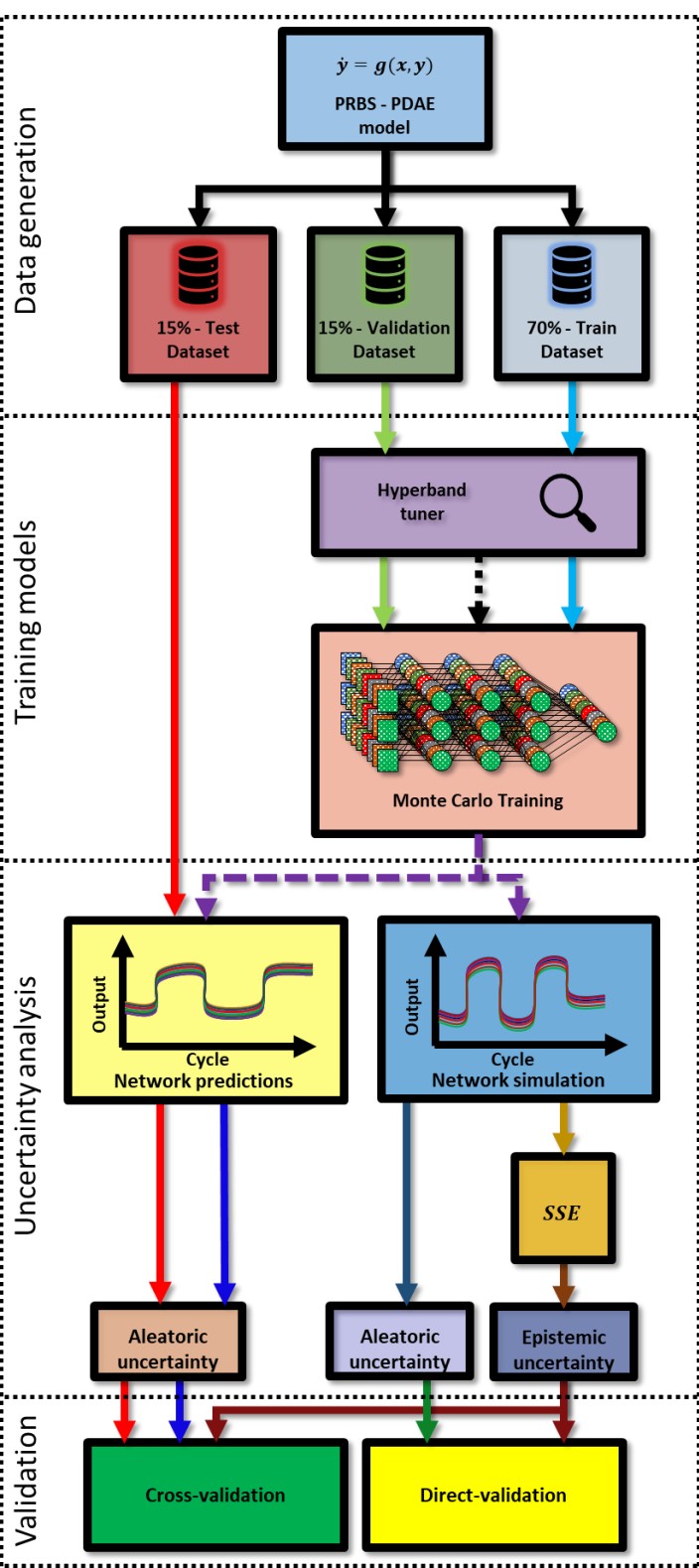

**Figure 1.** Proposed methodology. Dashed lines are models or configurations, and solid lines are data.

The first step is to obtain the data to identify the artificial intelligence models. The data source can be of two distinct origins: experimental or synthetic. This work uses synthetic data extracted from a partial differential algebraic equation (PDAE) model, a rigorous model of the pressure swing adsorption (PSA) unit implemented in gPROMS and used as a virtual plant. If one is interested in reproducing this model, please consult

Silvas' and Regufe's works [18,19]. The generated data is divided into three subsets to train, validate, and test. The generated data is divided into three subsets to train, validate and test. Following the diagram provided by Figure 1, the next step is to train the models. This starts with the definition of the artificial neural network (ANN) architecture and training parameters, the so-called hyperparameters. In this way, a hyperspace needs to be defined, where a proper optimization technique can be applied to identify the optimal set of hyperparameters within this space. The model's hyperparameters were optimized by the hyperband method proposed by O'Malley et al., 2019 [20].

This step is crucial to obtain a reference architecture to the Monte Carlo uncertainties training (MCUT) process. Once an optimal set of hyperparameters are defined, the MCUT process is performed. This is done over the training step, where the ANN parameters are estimated. The internal ANN parameters are the weights and bias. The number of these internal parameters is defined once the architecture is defined. However, repeating the deep feedforward neural network (DFNN) learning step, several model-fitting options can be found with equivalent performance. The MCUT will generate a set of models that represents the reference model found by the Hyperband method but with different internal parameters.

The next step of the methodology is uncertainty analysis. The main idea of this step is to evaluate the two components of uncertainty. Abdar et al., 2021 [15] divide the uncertainty components into two: epistemic uncertainty and aleatory uncertainty. The epistemic uncertainty is associated with the model distribution over the model parameters. On the other hand, the aleatoric uncertainty is related to the data variability [15]. Therefore, considering that the previously identified models represent the same model, these uncertainty components are evaluated. The validation of the training with the uncertainty assessment of the DFNN models is the final step of the proposed methodology.

## 3. Case Study

The Fischer–Tropsch process converts syngas into a complex mixture of hydrocarbons and oxygenated compounds, such as methanol or synthetic fuel, with a higher added value. Usually, a preliminary purification step is necessary to remove impurities and adjust the composition to the values specified for the process. A technological alternative that has attracted interest in recent years is adsorption-based separation. An example is the pressure swing adsorption (PSA) which presents a low cost of installation/operation and high efficiency while also associating the ability to achieve high levels of purity and recovery, flexibility, and simplicity of construction and operation [21–24].

This work uses a case study where a PSA unit purifies syngas. The ratio between $H_2$ and CO productivity is an important performance parameter due to the Fisher–Tropsh application. However, the measurement of concentrations is usually related to a high measurement of deadtime. The measurement deadtime problem is generally addressed through soft sensors. On the other hand, accessing these variables is subject to uncertainties.

The syngas purification process used in this work was proposed in Regufe et al., 2015 [19]. The process was designed with five steps: co-current pressurization, feed, rinse, blowdown, and purge, as shown in Figure 2.

Figure 2 shows that the products are obtained in a stream enriched in $CO_2$ and another rich in $H_2$ and CO. This last steam feeds the Fischer–Tropsch process with a stoichiometric $H_2/CO$ equal to 2.3. Regufe et al., 2015 [19] present the adopted premises and the mathematical model of the process, which is a nonlinear partial differential equations (PDEs) model that characterizes the system's mass, momentum, and energy balances. For more details about the process model, please consult Silva et al., 1999 and Regufe et al., 2015 [18,19].

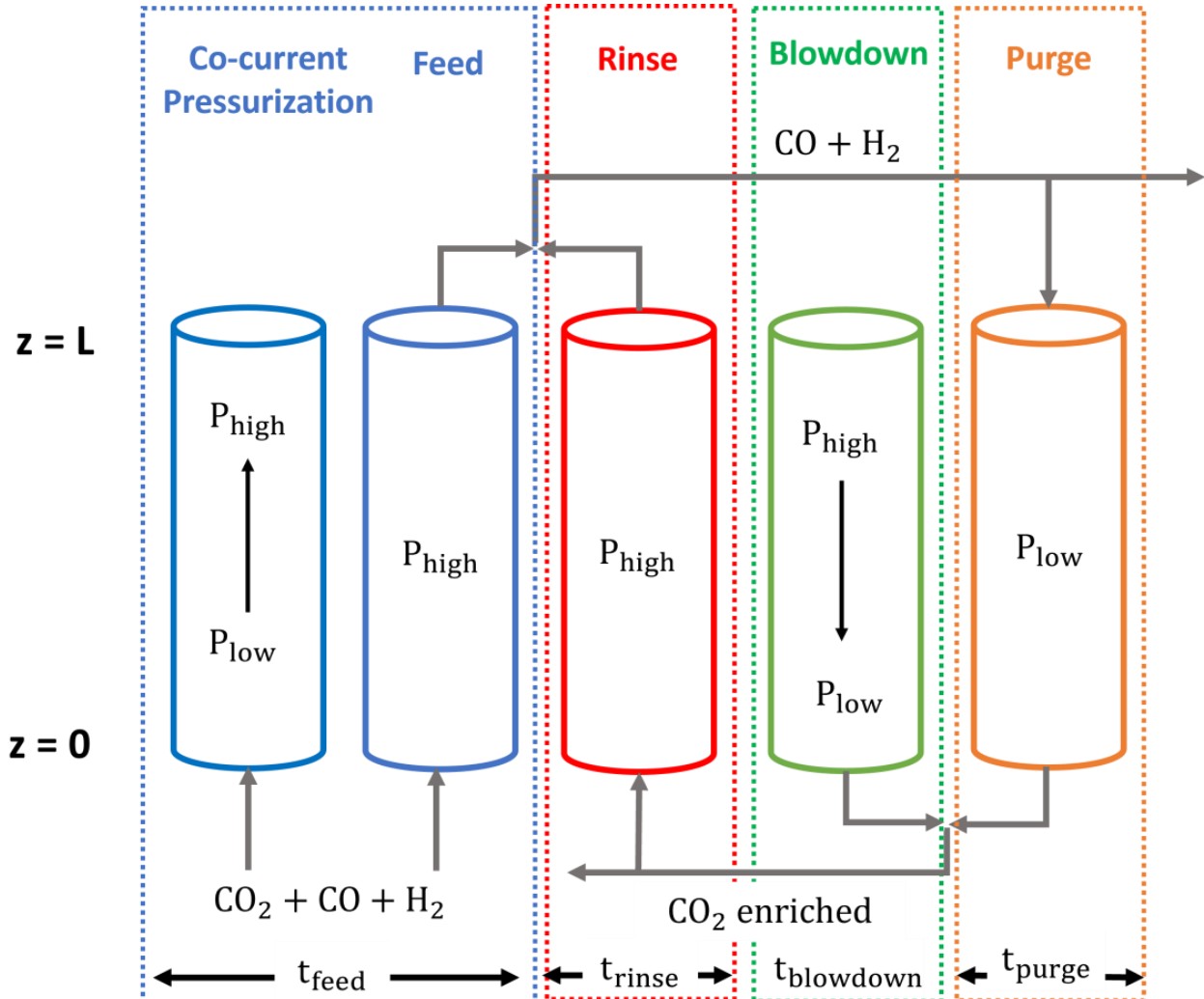

**Figure 2.** Cycle steps in a pressure swing adsorption unit for the syngas.

### 3.1. Data Acquisition

A software-in-the-loop (SIL) approach was applied to access the PSA phenomenological model to generate the synthetic data set for the training, test, and validation. The quality of the trained machine learning models will depend on the quality and volume of data provided in the training, validation, and testing stages. In complex systems, such as PSA, running long experimental tests to obtain a significant amount of data is difficult. It is important to highlight that industrial data plays an important role in soft sensors. On the other hand, there are variables that are difficult to be measured, or their measurement is expansive. In this situation, synthetic data is an alternative approach. This article uses synthetic data provided by a rigorous mechanistic simulator. To obtain predictions of variables that are difficult to measure in a pressure swing adsorption unit. Then, the synthetic data is encoded into a computationally light model used as a soft sensor. A new paragraph has been added to clarify this point further. In this way, this work obtains a set of data through a phenomenological model experimentally validated to overcome the difficulty of obtaining experimental data.

Additionally, it is essential to ensure that the data collected is representative of the system's behavior. In this way, a pseudo-random binary sequence (PRBS) signal was generated and inputted in the virtual plant to disturb the system and create the dataset.

This disturbance was done in the PSA input variables within a range listed in Table 1. The minimum and maximum values presented in Table 1 are related to the unit operation conditions referred in Nogueira et al., 2020 [25].

**Table 1.** Reference values for the process variables.

|  | $t_{feed}$/(s) | $t_{purge}$/(s) | $t_{rinse}$/(s) | $P_{high}$/(bar) | $P_{low}$/(bar) | $Q_{rinse}$/(SLPM) | $Q_{purge}$/(SLPM) | $T_{inlet}$/(K) |
|---|---|---|---|---|---|---|---|---|
| Minimum | 380 | 80 | 187 | 3.4 | 0.55 | 0.425 | 0.225 | 304 |
| Maximum | 680 | 110 | 253 | 5.0 | 1.10 | 0.575 | 0.345 | 350 |

The final dataset has 25,050 points containing information about the process dynamics. Figure 3 shows the first 500 cycles of the generated inputs in a normalized form. The expected random behavior of a PRBS signal is seen in Figure 3.

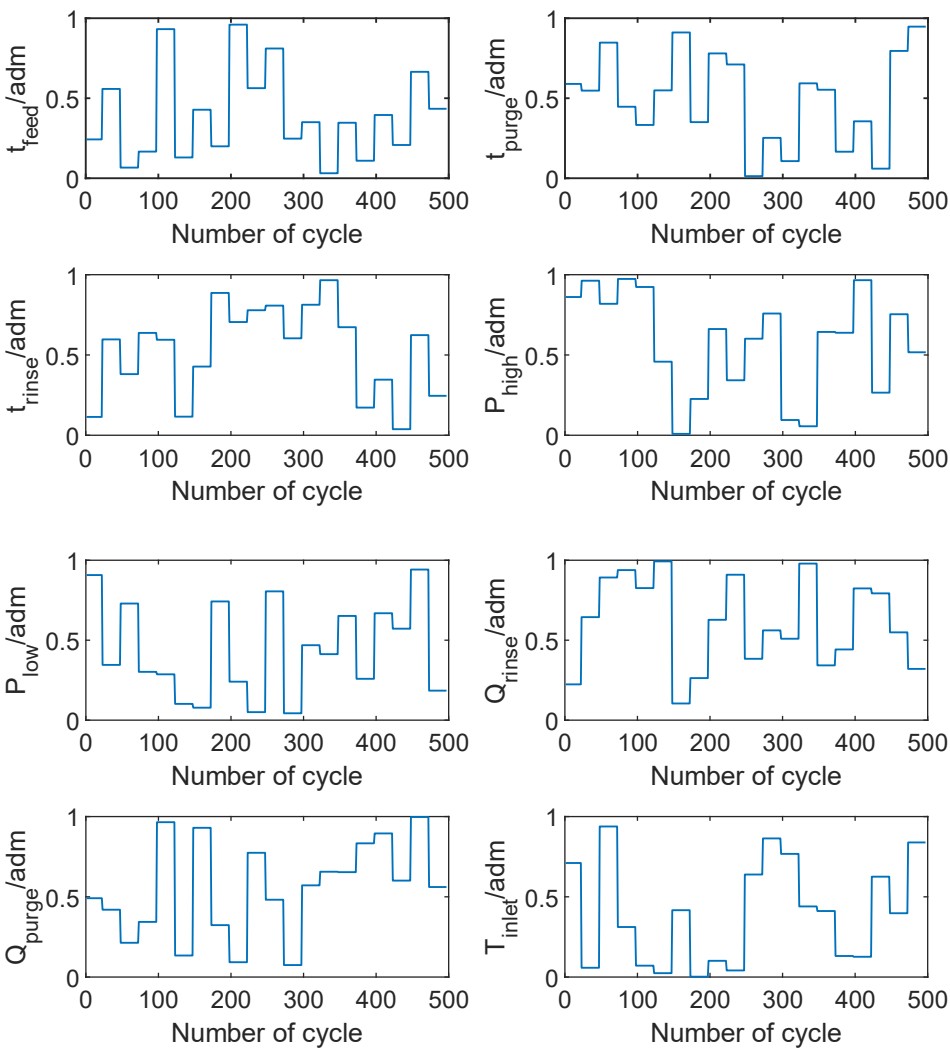

**Figure 3.** PRBS inputs signals.

Additionally, these input variables must not be correlated. This allows for no deviations in the data and, consequently, training, validation, and test will not present undesirable tendencies. This correlation analysis is performed in Figure 4, in which all inputs variables have the respective correlation presented in a heat map. It is possible to see that all inputs signals have a near-zero correlation coefficient. This indicates the input space given by the latin-hypercube sample (LHS) algorithm was well designed—i.e., significant

coverage of the area without creating unintended cross-correlation. These disturb signals included in the system are generated with LHS precisely to minimize the existence of a correlation between the input data. It is important to note that this principle of the design of experiments area should always be applied when performing an experiment, whether the data source is real or not. Otherwise, the input correlations can mask behaviors that need to be detected to obtain a proper model. The reviewer's point of view is applicable if the data is not collected from an experiment. Its source is historical data.

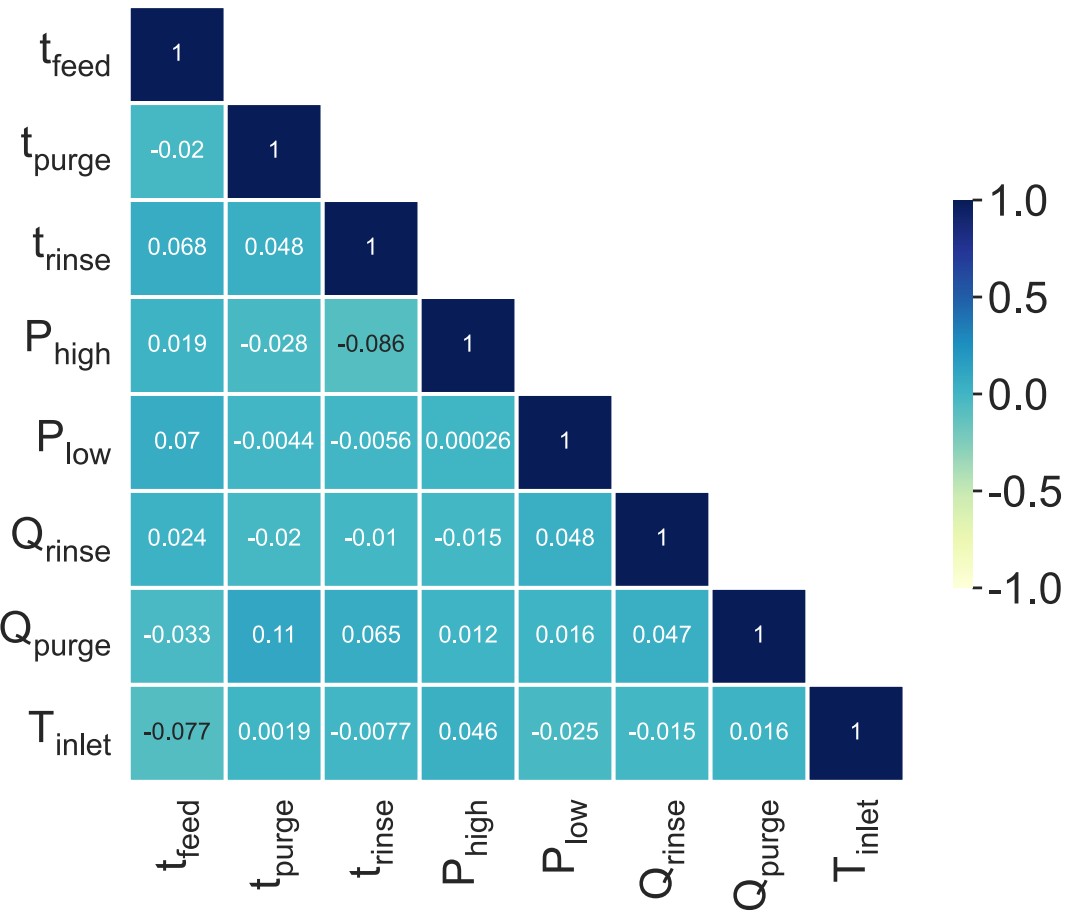

**Figure 4.** Correlation heatmap of the PRBS inputs signals.

### 3.2. Predictor and Data Structure

The second step of the proposed methodology, presented in Figure 1, starts by defining the predictor to be used. This work uses a nonlinear autoregressive exogenous model (NARX) to represent the $H_2/CO$ productivity of the PSA plant. The proposed predictor is written as

$$y_{k+1} = F(y_k, y_{k-1}, \ldots, y_{k-na}, u_k, u_{k-1}, \ldots, u_{k-nb}) \tag{1}$$

where $y$ represents the output model in each $k$ time step, and $u$ represents the vector of model inputs in each time step, $na$ and $nb$ are the input and regressor order, respectively. The Lipschitz index usually defines these values. Figure 5 presents the Lipschitz index for the data collected. As it is possible to see from this figure, the optimal regressor order is $na = 3$ for the input variables and $nb = 1$ for the outputs.

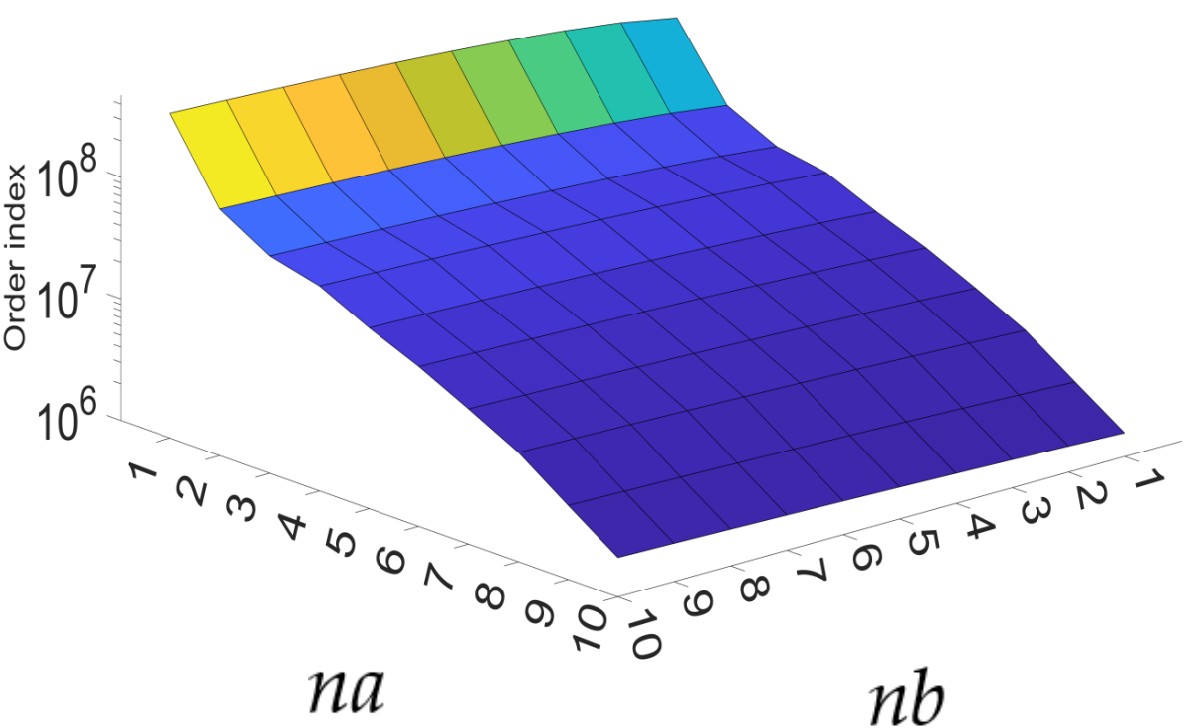

**Figure 5.** Lipschitz index for the inputs of the $H_2/CO$ productivity model.

### 3.3. Hyperparameter Tuning—Hyperband

Hyperparameters are variables that control the training process of artificial neural networks. They affect the final model performance significantly and must be selected carefully. The literature usually splits them into two categories: model and algorithmic hyperparameters. The first affects the model topology, e.g., number of layers, number of neurons, activation functions, and more, while the latter modifies the training algorithm parameters—learning rate policy, momentum, dropout rate, number of epochs, batch size, among others. However, the increasing complexity of available models and algorithms leads to many possibilities, making this selection a non-trivial task.

Machine learning practitioners usually employ significant computational power methods to find the best combination of hyperparameters, e.g., random grid search. This method lists all possible combinations of hyperparameters and randomly samples a small portion of configurations to train (serially or in parallel); then, the model with the best performance on a separate data set is selected. This is very time-demanding since it requires the complete training of all models to select the best configuration.

Recently, Li et al., 2018 [26] introduced the Hyperband algorithm, which improved the search speed up to 30 × compared to random grid search in benchmark problems. Hyperband formulates hyperparameter optimization as a pure-exploration problem where a predefined resource is allocated to randomly sampled configurations. Then, only the most promising configurations are given more resources (epochs). Hyperband requires two input parameters: the maximum amount of resources (epochs) and the proportion of configurations discarded in each round of successive halving (factor). In this case study, a maximum of 150 epochs and a factor of 4 were used, and the hyperspace and the results are summarized in Table 2. In general, this step uses fewer epochs to allow the method to explore a wider region of the hyperspace. This is a usual approach in the

literature [9,13,26–28]. Once the hyperparameters are defined, the epochs are increased for the training of the final structure.

**Table 2.** Hyperparameter search space and results for DFNN.

| | Hyperparameters of DFNN | |
| --- | --- | --- |
| | **Hyperspace** | **Results** |
| Initial learning rate | $\{1 \times 10^{-4}, 1 \times 10^{-3}, 1 \times 10^{-1}\}$ | $\{1 \times 10^{-2}\}$ |
| Number of dense layers | {1, 2, 3, 4, 5} | {3} |
| Recurrent layer type | - | |
| Number of neurons in the recurrent layers | 50 to 180, every 20 | 90 |
| Activation function in the recurrent layers | {relu, tanh} | {relu} |

On the other hand, it is necessary to ensure that the model used is suitable for what is desired. In this sense, several works in the literature have pointed to the deep neural network as the most suitable machine learning solution to model complex dynamic systems [10,13,29]. For instance, Rebello et al., 2022 [13] and Oliveira et al., 2020 [10] compared several machine learning approaches, concluding that deep learning was able to better describe the dynamics of a pressure swing adsorption unit. Schweidtmann et al., 2021 [29] points out that among ML techniques—such as random forests, support vector machines, spline functions, among others—deep learning is the most suitable for learning complex dependencies.

In this way, Figure 6 provides a brief comparison between deep learning and two other strategies. Feedforward neural networks and recurrent neural networks were identified to perform this comparison. These models were identified following the procedure described above. Thus, the FNN and RNN optimal structures obtained are described in Table 3. Therefore, using the parameters indicated in Table 3, one can reproduce these models.

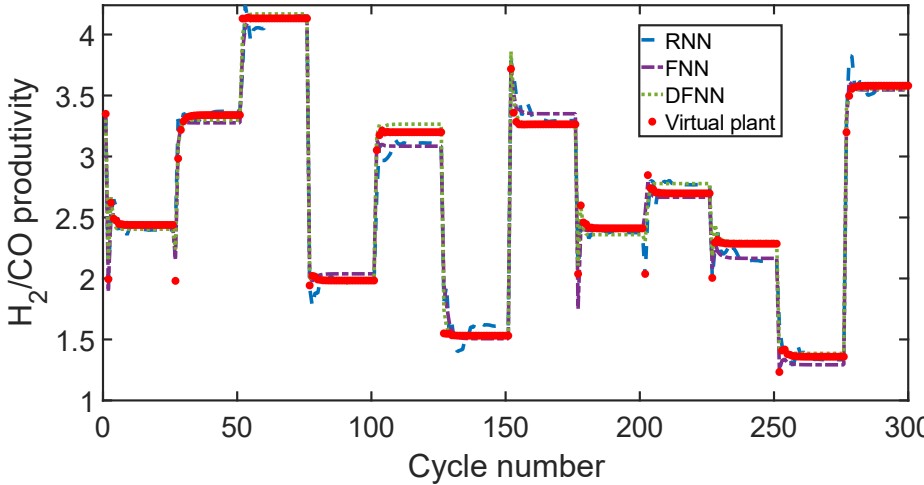

**Figure 6.** Comparison of different network architectures for the $H_2$/CO productivity model.

**Table 3.** Architectures of the different $H_2$ and CO productivity model.

| | **RNN** | **FNN** |
| --- | --- | --- |
| Initial learning rate | $\{1 \times 10^{-3}\}$ | $\{1 \times 10^{-3}\}$ |
| Number of layers | {5} | {1} |
| Number of neurons of the layers | {100, 60, 100, 40, 60} | {150} |
| Activation function of the layers | {tanh, tanh, relu, relu, tanh} | {relu} |

Additionally, through the mean squared error (MSE) and mean absolute error (MAE) values presented in Table 4, it is possible to verify that the DFNN model fits better and adequately represents the PSA system. From these results, it is clear that the DNN is the most suitable approach in the present case. This is in line with the literature described above.

**Table 4.** Adjustment indices of the different $H_2CO$ productivity models.

| Network | MAE | MSE |
| --- | --- | --- |
| RNN | 0.5688 | 0.3621 |
| FNN | 0.2209 | 0.0796 |
| DFNN | 0.1746 | 0.0587 |

### 3.4. Monte Carlo Training

The Monte Carlo simulation is a versatile tool used in several applications. In this work, the Monte Carlo's purpose is to evaluate the uncertainty of the soft sensor developed for the PSA unit. This work proposes to use the law of propagation of PDFs proposed by BIPM et al., 2008 [30] to train several models with the same architecture defined by the Hyperband method. This allows accessing the empirical model uncertainty. Each train will lead to a given set of DFNN parameters that yield a satisfactory model. Thus, the MC sorts samples of these parameters and trains a new model for each sample. This process is here called Monte Carlo uncertainties training (MCUT). The training epochs, batch size, and learning rate were sorted by a randomly uniform distribution from 300 to 350, 32 to 128, and 0.009 to 0.015.

An early stopping option was adopted in the algorithm to reduce the computational effort during the MCUT. Therefore, the training is interrupted with the patience option activated on TensorFlow following the fitting performance variable, MAE or MSE. The training stops if these variables do not improve within the stipulated epoch. The patience was set to 100 epochs for this step of the work. That is, if, in 100 epochs, the MAE value does not improve, the training is stopped. Figure 7 shows the histograms of the training indexes, epoch effectively trained, and the training duration for the MCUT proposed here.

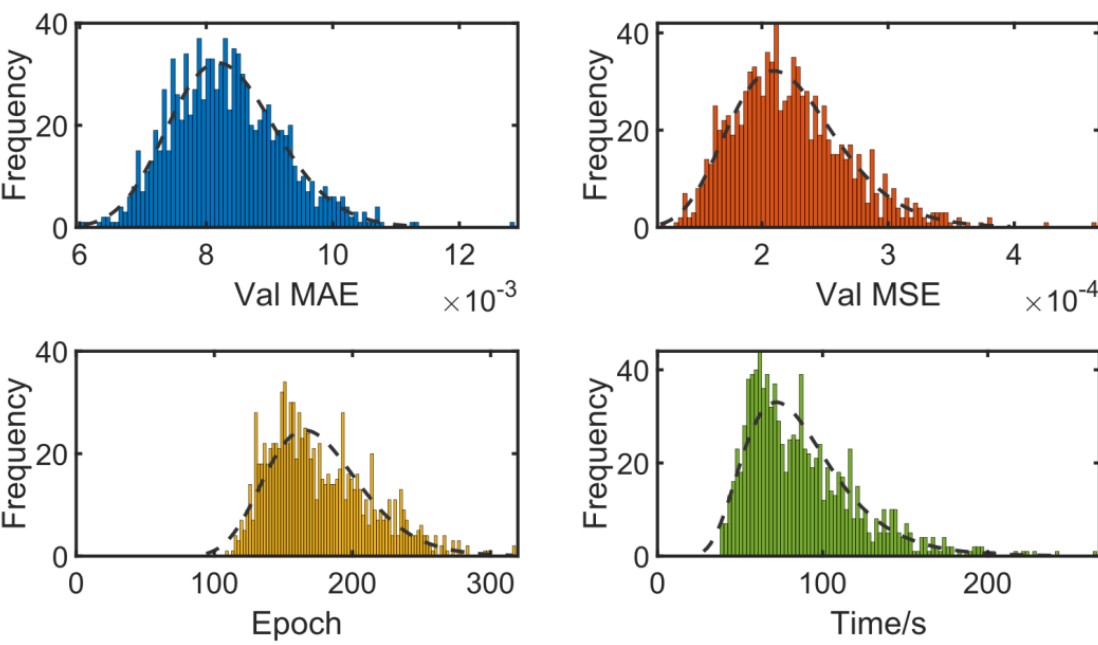

**Figure 7.** Histogram of the training index.

Figure 7 also shows the MAE and MSE histogram for the MCT process. These values were calculated for each trained network with the test dataset, typically, MAE values less than $1 \times 10^{-2}$ ensure a suitable fitting. To ensure a good model fitting, a final analysis is performed using the parity plot in Figure 8. Since 1000 different models were evaluated, the amount of information available allows us to assess how the uncertainty of the model interferes in the parity plot. Thus, the minimum, maximum, and most probable values were plotted for all trained models. Considering the quantiles representing the probability of $p$ = [0.025, 0.5, 0.995], Figure 8, green values represent the minimum values of the quantile, the maximum values are in cyan, the most probable value in blue, and the red line is the reference bisector. As it is possible to see, the density of points is higher in the diagonal line, indicating that the predictions agree with the actual states. The observed deviations are computed in the uncertainty of the soft sensor, incorporating it in the sensor prediction.

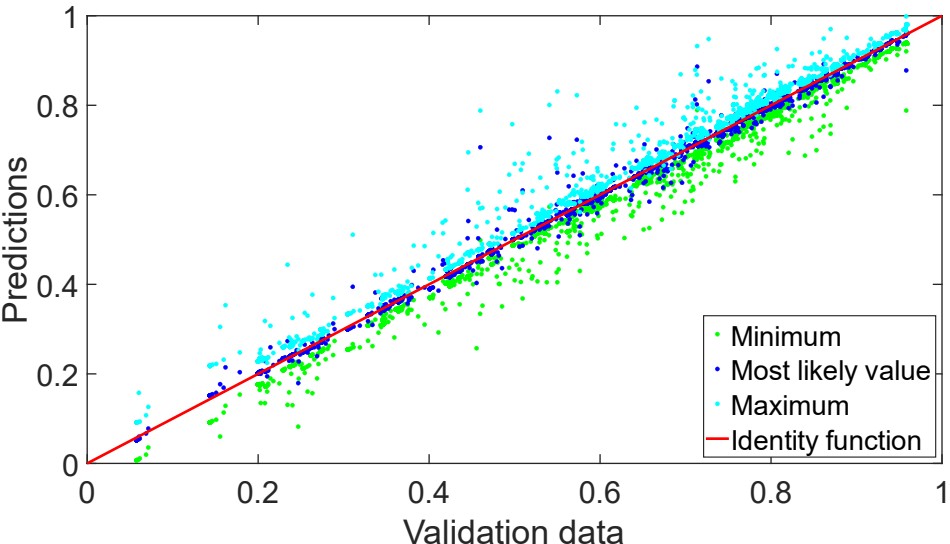

**Figure 8.** Parity plot of the validation versus model predictions.

### 3.5. Uncertainty Analysis

Uncertainty analysis of trained models is performed in two steps. First is identifying the epistemic uncertainty, which characterizes the uncertainty of the model itself. One way to estimate its value is to assume that the variance of the model's output follows an inverse gamma distribution with shape parameters $a$ and scale $b$, as proposed by References [31,32]

$$\sigma \sim \Gamma^{-1}(a, b) \tag{2}$$

where $\sigma$ is the variance of the model.

Gelman et al., 2013 [33] propose to assume that the shape parameter equals the mean between the number of prior information and the data. In the case of this work, it is assumed that the MCUT has non-previous information because no data about the model uncertainty is available. These assumptions imply in to obtain the shape parameter as follow

$$a = \frac{N_{data}}{2} \tag{3}$$

In its turn, the scale parameter $b$, based on the above assumption, is obtained by Equation (4) as a function of the sum of squared errors (SSE) calculated during the MCUT. The SSE value is obtained with the data train. This work assumes that MCUT provides a set of PDFs of the parameters of the DFNN models as

$$b = \frac{2}{\text{SSE}} = \frac{2}{\Sigma \left( y^m - y^d \right)^2} \tag{4}$$

where $y^m$ is the predicted output and $y^d$ is the data output.

Assuming the previous hypothesis and equations, Figure 9 shows the SSE and $\sigma$ histogram for all trained models, adjusted to a lognormal distribution.

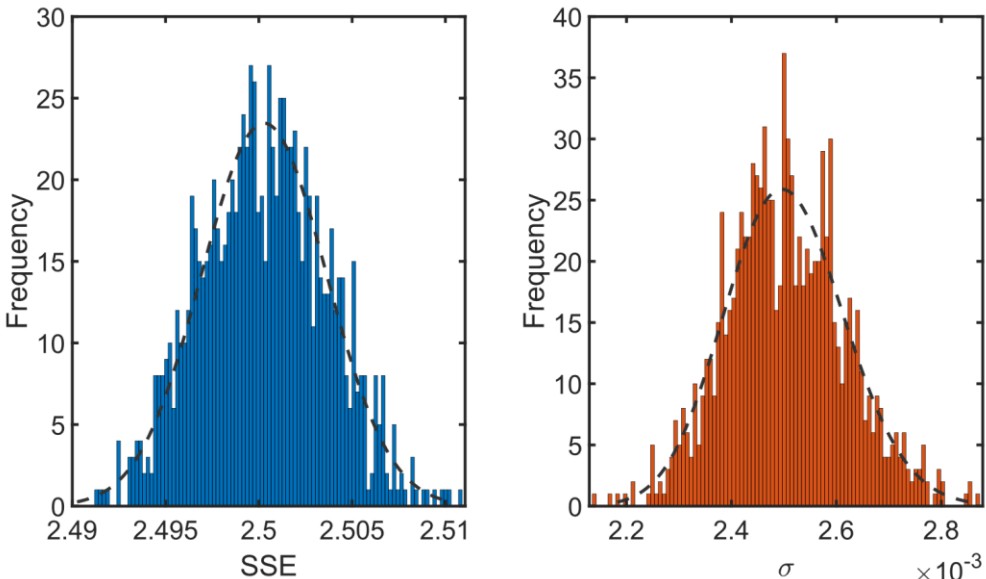

**Figure 9.** Histograms of the SSE and DFNN variance.

On the other hand, a second step is to compute the random uncertainty. This arises from the randomness of training and network prediction. However, it is enough for this analysis to obtain the confidence limits by calculating the quantiles for each predicted cycle. The random uncertainty can be briefly seen in Figure 10. The figure compares the DFNNs predictions for 400 cycles of 100 sampled virtual plant output. It is possible to see the variation of the models compared to the PDAE solution (dotted line). The variation observed in Figure 10 is consistent with the distribution of the MSE and MAE shown in Figure 7, as better is the model fit, the smaller the value of the MSE and MAE will be and, consequently, the smaller will be the distance between the prediction of the DFNN model and the virtual plant.

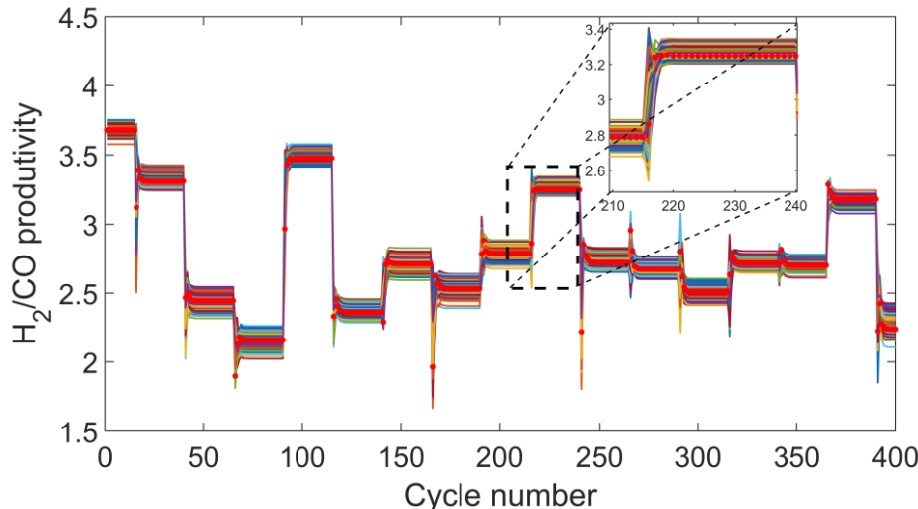

**Figure 10.** Prediction for the test dataset compared with the virtual plant.

Hence, based on the steps mentioned above, the prediction uncertainty is computed. Then, it is possible to evaluate the final DFNN prediction uncertainty for the test data with the virtual plant solution. Figure 11 portrays this evaluation, presenting the virtual plant actual states, the two uncertainty sources of the DFNN, and the corresponding most likely value (MLV). It is possible to see in Figure 10 that the MLV and the virtual plant states are close. Additionally, the aleatoric uncertainty is the green region around the MLV prediction and has a minor contribution. The epistemic uncertainty, on the other hand, is the gray region. This uncertainty contribution is more significant than the aleatoric uncertainty because it brings together the contributions of several model parameters. The epistemic uncertainty is obtained through the prediction for each cycle of the test data to all models trained by the MCUT. Then, each cycle of each model is summed with one sampled variance from the inverse-gamma distribution of Equation (2). With all response curves with the associated uncertainty, the most likely value (MLV) and the limits are obtained assuming the desired probability is $p$ = [0.005, 0.5, 0.995]. It is possible to see that the epistemic uncertainty contribution is more significant than the aleatoric as expected because they consider more sources of uncertainty.

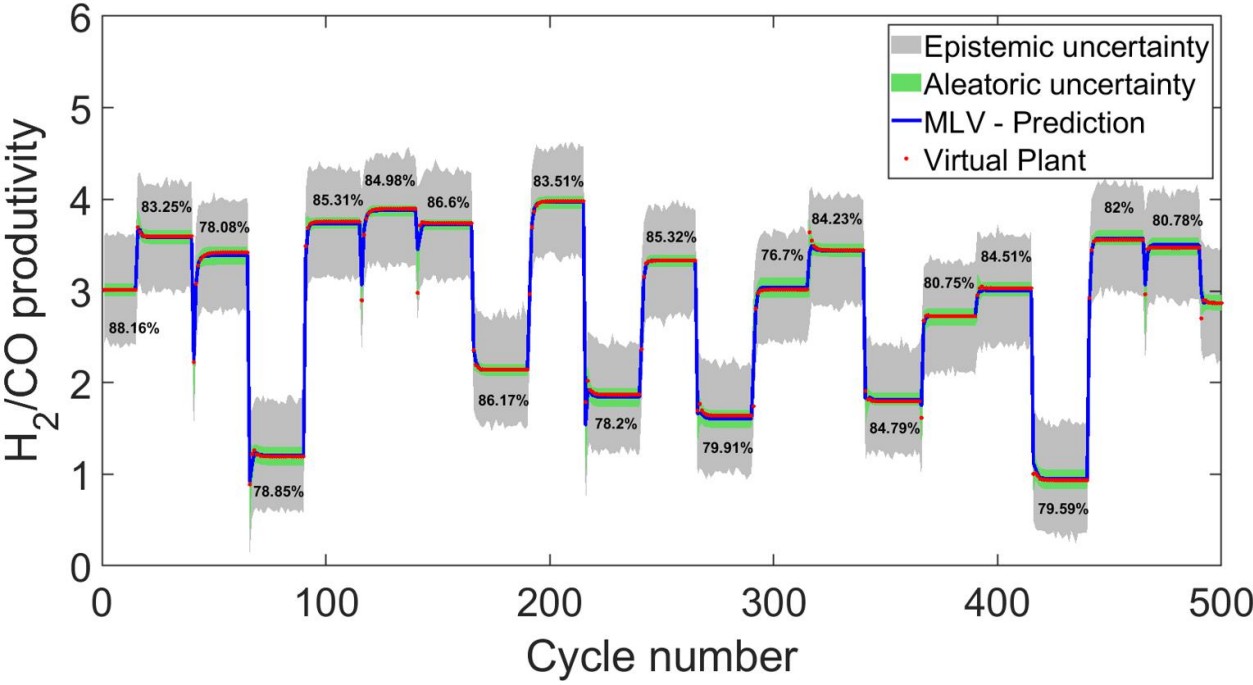

**Figure 11.** DFNN prediction and uncertainties compared with the virtual plant and relative uncertainties.

Epistemic uncertainty is influenced by several parameters of the DFNN model, which makes it considerably more significant than random uncertainty. Figure 11 shows the mean relative values of 25 cycles of the model uncertainties and the epistemic percentual at a given cyclic steady state. It is possible to see that epistemic uncertainty impacts greater than 75% on the general uncertainty.

Epistemic uncertainty is inherent in building a dynamic model with many parameters. In general, it can only be reduced if some of the sources are fully understood or are disregarded in the calculation. In this work, it was considered that uncertainty is associated with the lack of knowledge of the true value for the number of trained epochs, batch size, and learning rate. Thus, one of the ways to reduce uncertainty is to guarantee (or assume as a hypothesis) that one of these parameters does not have uncertainty or that it is irrelevant concerning the others. However, it is considered that knowing the uncertainty of a model does not mean that the model is wrong. Still, it is understood that the true value of its prediction is within the coverage region.

From the above results, it is possible to see that the MCUT proposed here yielded a virtual analyzer capable of predicting the process's leading property and providing its prediction uncertainty. In this way, it was possible to identify a more reliable model through a few extra steps in the model identification. On the other hand, the MCUT might be computationally exhaustive due to the identification of thousands of models. However, this step is done offline; therefore, the identification time is not a limiting factor. Furthermore, the final model predicts the state in real-time and their uncertainties, which compensates for the extra effort introduced to the model identification.

## 4. Conclusions

This work addresses the development of soft sensor for a chemical process based on deep feedforward neural networks. The uncertainty evaluation of the deep learning model is an open issue in the literature. It needs to be addressed to explore the potential of this technique in fields such as sensors development. Hence, this work proposed a methodology to evaluate the DFNN uncertainty based on a proposed Monte Carlo uncertainty training process.

A pressure swing adsorption unit for syngas separation was presented as a case study. The real-time measurement of the unit productivity is an important point of this process. However, this process presents a complex dynamic and a heavy phenomenological model. Therefore, the soft sensor is a good alternative to address the lack of online information.

In this way, the methodology proposed here was applied to develop an uncertainty-oriented soft sensor for real-time prediction of the PSA $H_2$-CO productivity. The proposed method made it possible to identify the two prediction uncertainty sources, the epistemic and the aleatory. The results prove that the Monte Carlo uncertainties training can yield a reliable model whose most provable value can follow the virtual plant tendency. At the same time, the uncertainty intervals are precisely presented. Therefore, it is possible to conclude that the proposed methodology can increase the reliability of the developed soft sensor without prejudice of the real-time capacities of the developed sensor. This provides further steps on applying deep learning techniques in soft sensors development.

**Author Contributions:** Conceptualization, I.B.R.N. and E.A.C.; Methodology, I.B.R.N., C.M.R. and E.A.C.; Writing—original draft preparation, I.B.R.N. and E.A.C.; Writing—review and editing, I.B.R.N., E.A.C., C.M.R., V.V.S. and A.M.R.; Supervision, I.B.R.N., A.E.R., A.M.R. and L.S. All authors have read and agreed to the published version of the manuscript.

**Funding:** This work was financially supported by: Project-NORTE-01-0145-FEDER-029384 funded by FEDER funds through NORTE 2020—Programa Operacional Regional do NORTE—and by national funds (PIDDAC) through FCT/MCTES. This work was also financially supported by: Base Funding-UIDB/50020/2020 of the Associate Laboratory LSRE-LCM—funded by national funds through FCT/MCTES (PIDDAC), Capes for its financial support, financial code 001 and FCT—Fundação para a Ciência e Tecnologia under CEEC Institucional program.

**Conflicts of Interest:** The authors declare no conflict of interest.

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
