# Peer review of "Mapping Uncertainties of Soft-Sensors Based on Deep Feedforward Neural Networks through a Novel Monte Carlo Uncertainties Training Process"

_processes, doi:10.3390/pr10020409_

Round 1
Reviewer 1 Report
Line 50: "Recently, the Soft Sensor literature has been pointing the issue of prediction" ... Citations are needed.
Line 72: Abbreviation PSA used without explanation.
Lines 178-179: Text unclear, does it mean that optimal na=1 and optimal nb=3?
Line 207: "...maximum of 150 epochs and a factor of 4 were..." - What was the reasoning behind the choice of these specific values?
Lines 218-219: "This idea advent of the fact that several parameters interfere in fitting DFNN models; likewise, learning rate, epochs, and batch size." - This needs to be modified, the sentence is not clear in its present form.
Lines 284-297: The obtained results of Epistemic and Aleatoric uncertainties should be expressed in quantitative terms, their graphical presentation in Fig. 10 is not sufficient. Furthermore, the relatively high values of Epistemic uncertainty should be discussed in more detail, especially the possibilities of its reduction.
General comment: This study is a purely simulation-based using a mathematical model of the process. This means, among other things, that it was possible to generate a relatively large amount of training/validation/testing data as needed, without any real limitations. It would be therefore appropriate to comment on the situation when a sufficiently accurate mathematical model would not be available for a given process and the necessary data would have to be generated experimentally. What is the minimum amount of data that would need to be generated experimentally for the proposed strategy? In other words - what are the limitations of the proposed strategy with regard to the amount of data required?
General comment: The manuscript needs to be proofread by a native English speaker.
Author Response
Thanks for your comments. Please see attached file.

Reviewer 2 Report
1.- Although it is true that Deep Learners have a difficulties with uncertainties, we need to point out that the use of Bayesian via Backpropagation has a long history all the way to 2016. For example you have
“Gal, Yarin, and Zoubin Ghahramani. "Dropout as a bayesian approximation: Representing model uncertainty in deep learning." international conference on machine learning. PMLR, 2016.”
by Ghahramani. Therefore a correction needs to be done in that statement.
2.- Where are the citations sustaining “Recently, the Soft Sensor literature has been pointing the issue of prediction uncertainty as an unsolved issue in this field.”
3.- Data is totally artificial, Did you ever try with real data? After all the business of soft sensors needs such data.
4 Given the complexities of the model, we would like a deeper explanation of the Monte Carlo method used in the paper for the Deep Learner. Mostly because at “3.4. Monte Carlo Training,” MC seems to be used for “sorting” parameters (We suppose you run MC and use the expected value for the parameter). Then, we suppose a back-propagation and stochastic gradient is used for the training under such parameters. However, for example, at “Deep learning-enhanced variational Monte Carlo method for quantum many-body physic” They describe a full algorithm for the Variational Monte Carlo in Deep Learners. Did you try to use something like that?
5.- Something we need to point out is that your scenario at the LHS does not have high correlation. However, this is not necessary the case in real life scenarios. We believe this is due to the use of artificial data, therefore the need to expand the study into real data.
6. Improve your Acronyms, there are several typos at them.
7. While looking at your uncertainty analysis, you use the inverse Gamma distribution to model the variance of the gamma of the model. Given your assumptions of a and b, Did you try to compute the confidence intervals for gamma? How they affect the estimation of the Epistemic uncertainty model? Are they related to the error between the output and test data?
8. While looking at the bottom figure page 11 (Over-imposing of all uncertainties), we noticed that the Deep Learner is working as a filter for Epi and Aleatory uncertainties to produce a better estimation. Did you try to see the effect of the Deep Learner as filter in your evaluation?
9. We feel that at the conclusion there is a need of hard data to sustain the work of using Deep Learners in Soft Sensor estimation (A deeper analysis is needed)
Author Response

(The authors gave the same response as above.)

Reviewer 3 Report
The paper is well-written and organized. The experimentation is well-explained too. However, in my opinion, it needs a little more work in some directions. First, preliminary concepts must be better explained (uncertainty, soft sensors, and so on); For example, what kind of uncertainties the authors are referring to: error in data, unknown information, imprecise information, fuzzy information, and so on ? It must be explained (an example in the introduction section is, to me, mandatory to explain uncertainties, soft sensors, what kind of information you are talking about); Soft sensors must be explained more (maybe a background sections is mandatory). Second, the introduction section is incomplete and the motivation for using DFNN is not well justified. On the other hand, in a paper of that nature a complete study demands a complete experimentation with more models (classical or alternative artificial neural networks and deep learning). In fact, as I said, in the introduction section is not well motivated why DFNN is used...with 7 authors! (it could be done in a reasonable time because author employ tensorflow library) in the paper, other neural networks could be checked, and a comparison between different kind of neural networks could be provide to the scientific community.
In the Abstract, the sentence: "A way to deal with this problem is by using soft sensors and evaluating they uncertainties' ' maybe is "their uncertainties??". Authors said: "uncertainty identification of DL problem is an open issue in the literature" but this affirmation needs a deeper discussion in the paper, for example: which lines of work exist for uncertainty identification of machine learning models?, I do not understand this part. The affirmation " precise predictions", is this affirmation correct???: " because: can a prediction be precise ? prediction has a probability? Don't you? a prediction never is a truth, I mean when a prediction is a truth, it is not a prediction anymore
About the introduction, it is not clear in some aspects...authors need to provide readers with a more clear and convincing discussion about why DFNN is used and also authors must use other neural networks in order to provide the community with a complete study about that.
Why is deep learning a powerful tool for the field of data driven modeling ? which are the reasons?...I mean machine learning is a powerful tool too, even artificial intelligence is a powerful tool too, ...which are the advantages of using DL models in comparison with classical ML models or alternative neural networks...for these reasons this paper needs more work because alternative (neural network) models of machine learning should be evaluated in order to answer this important questions...The study by using a only one kind of Neural Network is, in my opinion, limited for a journal publication
Authors said: "A usual problem found in engineering is the measurement of unmeasurable quantities" but machine learning models are used to solve "the measurement of unmeasurable quantities"...I do not understand, because machine learning models are used to predict or classify something
In conclusion, improve the study in three lines: justification, introduction and experimentation.
Author Response

(The authors gave the same response as above.)

Round 2
Reviewer 2 Report
Given the extensive review, please only do a English text editing only to have a good quality paper. Thanks
Author Response
Thank you very much. We reviewed the manuscript language.
Reviewer 3 Report
Most of my comments have been addressed. However, I still have some doubts regarding the work and the correct replication of the results by other colleagues.
Authors said: "Please note that these are well-known concepts."
I know, but a scientific paper must be always self contained
Authors said: "We have added the requested comparison between different ML techniques and the details explaining the employment of DNN tools."
How this comparison have been performed?, because authors do not provide readers wih a public web where data, models and software are available. I mean, as far as I know the most of the articles that use machine learning models create a public web page where the models place the data and all the information necessary to be able to replicate the experiments. I recommend to the authors put their models, data and results on a public web page in order to other scientists can replicate the results shown in this paper.
In fact, I disagree with this affirmation: "Additionally, the TensorFlow library is a secondary element in this work." ...You are using a "virtual plant", data, learning machine models, so software, models, data, so on, they also an important part of this research
On the other hand, some affirmations need to be supported by the literature. For example: "Hence, the real-time evaluation of these variables associated with assessing their uncertainty is an important issue to be addressed. However, in soft sensing based on AI literature, it is not usual to address both perspectives simultaneously. This is the main contribution of this work; it presents a methodology capable of providing a comprehensive view of these problems." ...seems opinions...
Author Response
Thank you. Please see attached file.
